# Tumor Suppressors Having Oncogenic Functions: The Double Agents

**DOI:** 10.3390/cells10010046

**Published:** 2020-12-31

**Authors:** Neerajana Datta, Shrabastee Chakraborty, Malini Basu, Mrinal K. Ghosh

**Affiliations:** 1Cancer Biology and Inflammatory Disorder Division, Council of Scientific and Industrial Research-Indian Institute of Chemical Biology (CSIR-IICB), TRUE Campus, CN-6, Sector–V, Salt Lake, Kolkata-700091 & 4, Raja S.C. Mullick Road, Jadavpur, Kolkata-700032, India; neerajana.datta@gmail.com (N.D.); shrabastee.chakraborty@gmail.com (S.C.); 2Department of Microbiology, Dhruba Chand Halder College, Dakshin Barasat, South 24 Paraganas, West Bengal PIN-743372, India; drmalini.basu@gmail.com

**Keywords:** tumor suppressor genes, Rb, PTEN, FOXO, PML, cancer

## Abstract

Cancer progression involves multiple genetic and epigenetic events, which involve gain-of-functions of oncogenes and loss-of-functions of tumor suppressor genes. Classical tumor suppressor genes are recessive in nature, anti-proliferative, and frequently found inactivated or mutated in cancers. However, extensive research over the last few years have elucidated that certain tumor suppressor genes do not conform to these standard definitions and might act as “double agents”, playing contrasting roles in vivo in cells, where either due to haploinsufficiency, epigenetic hypermethylation, or due to involvement with multiple genetic and oncogenic events, they play an enhanced proliferative role and facilitate the pathogenesis of cancer. This review discusses and highlights some of these exceptions; the genetic events, cellular contexts, and mechanisms by which four important tumor suppressors—pRb, PTEN, FOXO, and PML display their oncogenic potentials and pro-survival traits in cancer.

## 1. Introduction

The concept of suppression of oncogenesis has well preceded the actual discovery of tumor suppressor genes. The seminal work by Henri Harris first highlighted the concept of tumor suppressor gene (TSG) [1]. He established that malignancy, a dominant character of tumor cells, can be suppressed in hybrid cells formed by the fusion of malignant and non-malignant “normal” cells, and that a loss of chromosome during segregation in these hybrids may result in reversion to malignancy. This indicates that the normal cells contributed a balancing factor to the hybrid cells to suppress their highly malignant character.

The concept of tumor suppression was further solidified by Alfred Knudson’s epidemiological study of retinoblastoma patients [2] that originated the “two-hit hypothesis”. He discussed that both the alleles of a gene need to be lost or mutated for the tumor to develop into a cancer. Further, TSGs in their potential capacity as gatekeepers and caretakers are responsible for suppressing “damaged” cells from developing oncogenic potential. Later, Haber and Harlow proposed a stricter definition of a TSG in that they are “genes that sustain loss of function mutations in the development of cancer” [3]. 

Thus was born the distinct two-box categorization of genes regulating oncogenesis: the oncogenes, resulting from the activation of proto-oncogenes, and the TSGs that act as brakes and help in monitoring and maintaining a controlled cell cycle. When seen from an implicit viewpoint, these two functions would be mutually exclusive for a single protein; however, emerging evidence makes the placement of some genes into one specific box difficult. Despite defining two antithetical gene categories in oncogenesis, some genes exhibit both oncogenic and tumor suppressive functions under specific cellular conditions. These “double agents” are slowly shifting the paradigm of cancer research by adding newer dimensions to erstwhile known perspectives. The postulates of Hanahan and Weinberg stating that all cancers must necessarily acquire ten “essential alterations in cell physiology that collectively dictate malignant growth” [4] could clarify these contradictions. This proposes that oncogenes and TSGs participate in a complex intracellular signaling circuitry, which can explain their diversions from their traditional role. A recent work by Shen et al. addresses this issue, bringing into spotlight the identification, characterization, and expression of the proto-oncogenes, which behave as double agents by having tumor suppressive functions [5].

In this review, we focus on the other half: the TSGs. Despite fulfilling the characteristics and signatures defining a classical TSG, certain genes have shown increasing evidence of having oncogenic potential. 

Certain wild type (wt) TSGs, functioning as transcription factors and kinases, are capable to turn oncogenic based on their context-dependent transcriptional activation/repression of target molecules, without involving any genetic modification. The other category of TSGs, having basic cellular role in protein–protein interactions, expresses pro-survival functions through unique protein binding partners and interactions. We have identified these candidate genes from the Tumor Suppressor Gene Database TSGene 2.0 (https://bioinfo.uth.edu/TSGene), which provides a “comprehensive resource for pan-cancer analysis of TSGs” and have corroborated that with extensive literature mining [6]. TSG2.0 identifies 73 unique TSGs that have been coupled with instances of oncogenic behavior (Table 1).

As a comprehensive review of all the TSGs would not be possible here, we have specifically focused on three of them, Retinoblastoma (*RB1*), Foxo family, and *PML*, based on their substantial literature evidence. Additionally, although *PTEN* does not feature in the annotated list, we observed instances where, through an extensive interplay between several signaling cascades, occasionally oncogenic roles of *PTEN* have been identified. We have thus added *PTEN* in our review of “double-agent TSGs”. We have tried to re-evaluate here the different cellular mechanisms under specific demands of the microenvironment modulating the functions of well-reported TSGs.

## 2. TSG Mutations and Cancer 

Research over the years has established several examples of a TSG that does not fit the typical classical behavior and demonstrates oncogenic potential (Figure 1).

The important question is what makes the classical TSG behave as an oncogene. It has long been established that cancer is an evolutionary process, quite similar to the evolution of species. Cancer cells always know how to find the best-fit method for their survival and maintenance. This evolutionary change frequently occurs at the genetic level, where tumors evolve by mutation and selection acting on specific cells. TSGs are recessive at the cellular level, which means that according to the “two-hit hypothesis”, inactivation of both alleles is required for a reversal of their function. Alternatively, TSGs can also be regulated by haploinsufficiency, whereby one functional allele of the gene is lost by mutation or deletion, while the other allele, although undisturbed in the wild type form, is incapable of executing normal physiological function– preventing abnormal, uncontrolled cell proliferation [8]. Some well-known examples would be *Cdkn1b*, *Dntf1*, *FHIT*, and *APC.* The third mechanism, known as Dominant-Negative (DN) mechanism, can be illustrated by *ATM*, where certain missense mutations, despite expressing stable proteins, lead to a much greater reduction in ATM response than truncating mutations. Breast cancer patients with DN missense *ATM* mutations show an increased risk of developing ataxia telangiectasia [9,10]. The final mechanism is where certain heterozygous mono-allelic mutations in classical TSGs would turn on their gain-of-function (GoF), thus promoting cancer. Tumor suppressor p53 with both the wild type alleles is an authentic TSG; however, cancer-associated GoF mutations transform p53 into a potent oncogene.

The mechanism and role of GoF *mut*-p53 has been discussed in great details in prominent articles and reviews, hence is excluded from this work [11,12,13]. Appendix A illustrates the primary mechanisms of GoF *mut*-p53 and their subsequent role in oncogenesis.

However, apart from GoF mutation, p53 can also be inactivated through protein-protein interactions, activation of negative regulators of p53, viral infection, promoter methylation, mRNA dysregulation-mediated expression loss, etc., detailed analysis of which is beyond the scope of the current review.

It is natural for a TSG with two inactivating mutations on both alleles to have an oncogenic outcome; however, the instances when a TSG has either one or two functional alleles but still behaves as an oncogene require special attention. In this review, we highlight these “special” instances. 

## 3. Retinoblastoma (RB1)

Retinoblastoma associated protein (pRb) is the first-ever TSG to be identified and studied. It is a cell cycle regulator, restricting normal cells in the G1 phase; however, its function is often inactivated in cancers [14]. In primary retinoblastoma, Rb promoter is under extensive hypermethylation that encompasses the core promoter as well as other CpG dinucleotides present in the retinoblastoma CpG island [15]. A study of tumorigenesis in Drosophila eyes suggests the involvement of Notch-delta pathway in this mechanism [16]. DNA hypermethylation causes downregulated *RB1* gene expression in these tumors. Apart from phosphorylation, it is regulated by several post translational modifications (PTMs); for example, E3 ligase such as MDM2 promotes degradation of pRb [17], whereas deubiquitinase HAUSP stabilizes it, and protects it from proteasomal degradation [18]. pRb (p105), the classical controller of E2F target genes involved in the cell cycle, is widely considered a proliferation inhibitor and is functionally compromised in many human tumors. This deregulation is mainly due to mutations in *RB1* itself or in its family members, such as p107 or p130. Secondly, mutations causing increased pRb phosphorylation or increased expression of viral oncoproteins that target and inhibit pRb can also be the factor behind pRb inactivation. However, studies, mostly in colorectal cancers, show that pRb is expressed in higher levels as compared to adjacent normal tissues [19,20], is rarely mutated, and *RB1* locus is often amplified. This provides an interesting viewpoint that even in absence of mutations, pRb can participate in cancer progression through protein-protein interaction.

### 3.1. pRb and Angiogenesis

pRb regulates transcriptional activity of angiogenesis-related factors such as VEGF [21], HIF1 [22], ID2 [23], Oct-1, and IL-8 [24]. pRb family proteins are required for endothelial cell differentiation, mobilization, and proper formation of blood vessels [25].

### 3.2. pRb and Cell Cycle

An early report illustrated that pRb positively regulates cyclin D1 at early G1 stage of cell cycle, further regulating cell cycle progression [26]. This was later validated in non-Hodgkin’s lymphoma and mantle cell lymphoma, with elevated pRb level correlating with that of cyclin D1 [27]. This contradicts the well-accepted understanding that cyclin D1: CDK4/6 hyperphosphorylates and inactivates pRb in G1 phase, thus contributing to cell cycle progression. In this context, however, inhibition of pRb phosphorylation was shown to increase the resistance of esophageal cancer cell lines towards chemotherapeutic drug 5-Fluorouracil [28]. Later, a contrasting study added that cyclin D1:CDK4/6 mono-phosphorylates pRb in early G1 stage [29], especially under external stimuli like DNA damage response, whereby active pRb represses E2F-mediated transcription of target genes, mediating cell cycle arrest. Additionally, in acute myeloid leukemia (AML), tumor cell supernatant represses the cell cycle entry of activated T cells by inhibiting the phosphorylation of CDK4/6 sites on pRb. This pRb-mediated G0 to G1 blockage in cell cycle progression induces T cell immunosuppression leading to immune evasion of tumors [30]. Down the G1 phase of cell cycle, cyclin E:CDK2 hyperphosphorylates pRb, inactivating it, and removing E2F from its clutches [31]. This seesaw interaction between pRb and cyclin D1 remains an intriguing puzzle needing further validation. 

### 3.3. Anti-Apoptotic Role of pRb

Considering that the central role of pRb is in controlling proliferation, one of the earliest findings reveal anti-apoptotic role of pRb. This came from studies on *RB1-*null mice where *RB1* loss was associated with apoptosis in the nervous system, lens, and skeletal muscles [32]. E2F1 seems to be a key factor responsible for this phenomenon. pRb loss leads to E2F1 release and activation, followed by activation of E2F1 target genes encoding proapoptotic proteins, such as p73, caspases, and APAF-1 [33]. Additionally, though, pRb is able to bind and inhibit proapoptotic factors other than E2F1 [32].

This pRb-mediated coordinated control of apoptosis and proliferation varies with cellular contexts. Reversible inhibition of pRb during cell cycle through phosphorylation that regulates cell proliferation is functionally different from the complete *RB1* loss that induces apoptosis in *RB1*-null mice. Mechanistically, *RB1* loss leads to de-suppression of E2F leading to activation of both cell cycle and apoptotic genes, unless being acted upon by survival pathways or abrogation of p53 pathways that protect cells from *RB1* loss induced apoptosis [33]. On the other hand, CDK-mediated phosphorylation of pRb causes release of most E2Fs that induce cell cycle gene transcription; however, a smaller fraction of intact and stable P-pRB1-E2F1 complexes persist in proliferating cells where they interact at the promoters of apoptotic genes thus repressing their expression [34,35]. Thus, pRb phosphorylation can provide both proliferative and survival advantages to cancer cells depending on the cellular context. Further, possibly due to direct inhibition of apoptotic genes, pRb ablation is seen to be associated with increased sensitivity to cell death; in parallel, pRb restoration inhibited apoptosis upon various apoptotic stimuli, such as ionizing radiation, p53 overexpression, ceramide, and IFN-γ [33].

### 3.4. pRb and E2F Signaling

The transcription factor E2F is a key downstream target of pRb. Genetic inactivation of pRb pathway due to mutation or alterations in any of the members leads to de-regulation of pRb/E2F pathway in human cancers. Based on the observation that colorectal carcinoma cells possess an intact and rarely mutated *RB1*, new information has arrived in the study of pRb/E2F and Wnt/β-catenin pathways [36]. In human Saos-2 cells (*RB1^−^/p53^−^*, with weak basal Wnt activity), E2F1 transcriptionally activates ICAT (inhibitor of β-catenin) that inhibits β-catenin/TCF-mediated transcription of target genes, such as *MYC*, leading to E2F1-induced apoptosis [37]. In lung carcinoma cells and murine cells, E2F-led repression of the β-catenin/TCF pathway was mediated by upregulation of the E2F target gene *Siah1* [38] and Axin 2 induction [39], respectively. Thus, parallel understanding can be drawn in colon carcinoma cells, whereby intact pRb hampers effective functioning of E2F1, preventing E2F1-mediated transcriptional repression of TCF4/β-catenin target genes, leading to an activated Wnt/β-catenin pathway. Additionally, E2F1 is inhibited by colorectal oncoprotein CDK8, which along with an intact pRb helps in enhancing the activity of β-catenin, thus increasing the expression of its key targets such as c-Myc [40].

Another interesting function of pRb is its interaction with anti-apoptotic factor BAG-1, driving the latter’s nuclear localization, thus providing a survival advantage in colorectal adenocarcinoma cells [41]. Furthermore, since BAG-1 promotes NF-κβ activation, pRb repression inhibits TNF-α-induced NF-κβ activation and cell survival [42]. Simultaneously it was observed that BAG-1 is a c-Myc transcriptional target, responsible for inhibiting c-Myc’s own intrinsic pro-apoptotic functions [43]. Hence, in colon cancer, pRb can play a critical role in inhibiting c-Myc-induced apoptosis by promoting pro-survival activity of BAG-1. The proliferative role played by pRb in colorectal cancer is illustrated in Figure 2.

### 3.5. pRb and Ras Families

Ras and pRb family members are implicated in the regulation of various cellular processes. Though pRb is primarily a tumor suppressor, deregulation of pRb pathway components, such as loss of the CDK inhibitor p16, amplification of Cyclin D1, or mutation of CDK4 is observed in a variety of tumors [44,45]. pRb is overexpressed in colon, thyroid, and head and neck carcinomas. Simultaneously activated mutations of the Ras family of proto-oncogenes are implicated in 30% of human cancers [46]. However, pRb-deficient tumors seem to lack activated Ras mutations. While one possible explanation is that they have increased levels of GTP-bound Ras that bypasses the requirement of Ras activation [47], the other explanation is that mutational Ras activation and transformation in *RB1*^−/−^ cells leads to a slower proliferation as observed in mouse embryonic fibroblasts, human osteosarcoma, and colorectal tumor cells [48]. On the contrary, cells lacking both p130 and p107, when transfected with H-Ras^V12^, show an increased proliferation [48]. This corroborates the physiological finding that colorectal cancer cells have high intact pRb, but low p107, owing to pRb-mediated transcriptional repression of p107. Thus, a functional *RB1* is required for the transformation activity of H-Ras^V12^, cellular proliferation, and anchorage-dependent growth in colorectal cells. These findings indicate an unrecognized oncogenic potential of tumor suppressor pRb and explain why human tumors do not show a simultaneous loss of pRb and activated Ras mutations. A recent study by Walter et al. further discusses the complicated crosstalk between the Ras family and pRB, whereby re-activating pRb in advanced tumors only transiently suppress cell proliferation due to an adaptive rewiring of MAPK signaling; in contrast, pRb helps modify the cell state thus reprogramming tumors towards a less metastatic stage [49].

Role of pRb as a proliferation marker is also implicated in embryogenesis, where pRb-deficient sensory neurons undergo incomplete differentiation and tumors undergo apoptosis [50,51]. Finally, activation of pRb in mice mammary glands causes precocious differentiation of mammary epithelial cells, suppresses cell proliferation, finally resulting in mammary adenocarcinoma [52]. Additionally, constitutive expression of pRb interferes with DNA synthesis and repair, resulting in an accumulation of damaged DNA and chromosomal abnormalities [52]. 

In conclusion, pRb’s role in controlling G1-S transition and E2F transcription factor activities during cell cycle progression has always been emphasized. Accompanied by Knudson’s two-hit hypothesis, this leads to pRb’s reputation as a tumor suppressor. However, recent evidences put pRb at the crossroads of multiple pathways, in a context-dependent manner, where intact *RB1* is often integral to promoting tumor initiation and progression. Most of this phenomenon is regulated by E2F1 where it plays a unique role of a conditional tumor suppressor inducing apoptosis as opposed to its more conventional role in G1/S transition. This paradoxical situation may restrict the use of certain CDK inhibitors as they can limit tumor growth as well as suppress apoptosis. 

## 4. PTEN

*PTEN* is a bona-fide TSG located on chromosome 10q23, a region commonly deleted or mutated in multiple human cancers. Somatic mutations of *PTEN*, identified in tumors of multiple histological origins, place it amongst the most commonly mutated genes in human cancer. Of interest, phosphorylation of C-terminus of PTEN (PTEN-CT) allows it to interact with its lipid anchoring C2 domain, thus conferring stability to PTEN, protecting it from E3 ligase mediated proteasomal degradation [53]. Therefore, naturally, mutations in PTEN-CT, as observed in several PTEN-associated cancers, are responsible for low PTEN expression [53,54]. 

### 4.1. Haploinsufficiency of PTEN

Germline mutations in *PTEN* cause embryonic lethality, altered germ layers, and anchorage independent growth, all hinting toward poor embryonic development [55]. Unsurprisingly, *PTEN* haploinsufficiency is responsible for the promotion of high-grade astrocyomas [56]. Additionally, Kwabi-Addo et al. observed that losing one wild type *PTEN* allele accelerated tumor progression as compared to mice containing two wild type alleles [57]. Deletion of at least one allele is observed in 60% of prostate cancer patients while a complete loss is linked to metastasis and androgen independent progression [58]. Thus, a dose-dependent inactivation of *PTEN* seems to dictate the rate of progression of prostate carcinoma.

### 4.2. Epigenetic Control of PTEN

The most common epigenetic means by which *PTEN* is controlled are promoter hypermethylation and histone deacetylation. Hypermethylation of CpG islands in *PTEN* promoter is found in thyroid cancers [59], sporadic breast cancer [60], hepatocellular carcinoma [61], melanoma [62], chronic myeloid leukemia (CML) [63]. In case of acute lymphoblastic leukemia (ALL) and CML, *PTEN* hypermethylation is also associated with chemoresistance [63,64]. In AML, PTEN activity is inhibited by histone deacetylation and histone methylation via the actions of transcription factors SAL4 [65] and Evi1 [66].

### 4.3. PTEN and PI3K-Akt/mTOR Pathway

The primary role of PTEN as a tumor suppressor is maintaining a tight regulation over cell proliferation, migration, survival, and angiogenesis by antagonizing the PI3K-Akt/mTOR pathway, a major signaling route in oncogenesis. Hence, it is expected that the inactivation of PTEN in human tumors should correlate with activation of the targeted PI3K-Akt/mTOR pathway and resultant tumorigenesis. However, a substantial set of renal cell carcinoma (RCC) patients harbor a co-existence of a high level of both PTEN and pAkt (active Akt) proteins, suggesting that Akt activation might occur independently of PTEN loss [67]. Similarly, while comparing the mutation status of activating PI3K (CA-PI3K) and PTEN level, a GoF of PTEN was observed from primary breast tumors to corresponding metastases [68]. It was also noted that PTEN loss and Akt inactivation are not perpetually interdependent neither obligatory for driving the metastatic process.

PI3K-Akt/mTOR pathway functions as a regulator of angiogenesis by modulating the expression of HIF-1α and HIF-2α [69,70]. However, in a subset of RCC patients with substantial PTEN expression, PTEN is seen to enhance HIF-2α activity, but not the protein level, by negative regulation of YY1, the HIF-2α co-repressor, and the downstream target genes MT1-MMP, in *VHL^−/−^* RCC cells [71]. This is an interesting tumor-promoting role of PTEN in early renal carcinogenesis. 

PTEN, via its antagonistic action on PI3K, elevates cellular Insulin Receptor Substrate-2 (IRS-2) level and enhances its interaction with PI3K. This PTEN-mediated feedback loop is likely an attempt by the cell to avoid growth inhibition and apoptosis [72]. In mice, ectopic expression of PTEN protein leads to enhanced metabolic phenotype [73], which is paradoxical as PTEN-loss is generally associated with increase in metabolic functions.

### 4.4. PTEN and p53 Interplay

In addition to its lipid phosphatase activity, PTEN shares some Akt-independent functions in regulating tumor suppression, such as its interaction with p53. PTEN and p53 together shoulder the bulk of tumor-suppressive support in a cancer cell, and they are the two most frequently mutated TSGs in human cancer [74]. There exists a multi-level co-operation between these two TSGs, whereby they enhance and prevent degradation of each other. PTEN prevents p53 from degradation by suppressing PI3K and thus inactivating p53 inhibitor Mdm2, and also by preventing Mdm2 transcription via binding to inhibitory phosphoinositol binding site [75,76]. However, PTEN also stimulates the stability of GoF *mut*-p53, enhancing the latter’s tumor promoting functions [77]. PTEN restoration in U373 and SNB19 glioblastoma (GBM) cells (both harboring p53 mutations R273H) leads to cellular proliferation as opposed to U87 and A172 GBM cell lines that possess a wt-p53. Mechanistically, PTEN forms a stable complex containing *mut*-p53, CREB-bind protein (CBP), and NFY resulting further transcriptional upregulation of oncogenes c-Myc and Bcl-XL causing increased cellular proliferation, survival and clonogenicity [78]. A clinical relevance was observed as GBM patients with *PTEN^+^/mut-p53^+^* tumors had a worse clinical outcome compared to *PTEN^-^/mut-p53^+^* tumors [77]. These works provide an explanation as to why simultaneous deletion of *PTEN* and mutation of p53 is rarely observed in human tumors. 

### 4.5. PTEN’s Dual Lipid and Protein Phosphatase Activity on Proliferation

Pnck, a calmodulin kinase highly overexpressed in renal and breast carcinoma, inhibits serum-induced ERK1/2 and p38 MAPK activity, inducing a strong proliferative capacity in the cells. Introduction of wt-PTEN or its protein phosphatase active but lipid phosphatase dead mutant enhances Pnck-mediated-ERK1/2 inhibition, causing further proliferation [79]. Cellular stress results in p38 MAPK-mediated induction of Pnck expression that further increases PTEN’s protein phosphatase activity. A slight decrease in Akt expression, but not its activity, reported in Pnck-expressing cells, could be a downstream result of Pnck-induced Hsp90 phosphorylation and inhibition-cum-degradation of Epidermal Growth Factor Receptor (EGFR) [80]. Well-established role of Akt in phosphorylation and inhibition of RAF-MEK-ERK pathway [81,82] questions this lowered expression. Since PI3K-Akt pathway is central to many PTEN-mediated oncogenic signaling pathways, further delineation is required for a detailed picture. Of particular importance could be PTEN’s role with special reference to Pnck in other axes of ERK signaling, namely, the Raf-Akt or EGFR-Akt crosstalks.

In addition, regulation of PTEN itself impacts the cellular processes involved. One such example is β-arrestin, the scaffold proteins that regulate subcellular distribution and differentially impact cellular proliferation and migration [83]. It enhances PTEN lipid phosphatase activity in negative regulation of PI3K-Akt activation and inhibition of cell proliferation. On the contrary, β-arrestin binds to the C2 domain of PTEN and modulates lipid phosphatase independent activity of PTEN by allowing the cells to retain the migratory potential and to overcome the inhibitory effects of PTEN. The central role of PTEN and PI3K-Akt signaling in rendering the cellular proliferation is illustrated in Figure 3. 

### 4.6. Immunosuppressive Function of PTEN in Tumor Microenvironment (TME)

PTEN-loss is generally associated with immunosuppressive tumor milieu. Constitutive PTEN-expression in the TME suppressed regulatory T cells and rendered Dendritic Cell (DC) and CD8+ T cells inactive, thus leading to an essentially immunosuppressive TME. Pharmacological inhibition of PTEN converted the Tregs into pro-inflammatory helper cells, activated the DC, and increased antigen presentation, resulting in a proinflammatory and immunogenic TME that aids rapid tumor regression [84]. Similarly in elder subjects, higher PTEN level in the DCs has been shown to decrease Akt-activation, antigen-uptake, and migration, contributing to immune suppression [85]. Upon siRNA-mediated PTEN silencing in DCs, DC survival and migration increased along with CD8+ expression [86]. 

### 4.7. Oncogenic Role of PTEN in Leukemia

PTEN is frequently deleted in T cell lineage ALL, where PTEN loss correlates with renewal of leukemia stem cells [87]. A contrasting role is observed in CML and B cell lineage (B-ALL)-like leukemia, where PTEN causes FOXO-dependent upregulation of p53 suppressor Bcl6, allowing ALL and CML cells to escape p53- and p21-mediated cellular senescence [88]. Unlike T-ALL, PTEN deletion here does not accelerate cell growth, rather has the opposite effect. In a subsequent study, the same group further identified pre-B-ALL as the only human cancer subtype lacking genetic lesions of PTEN [89].

During the development of mature B-cells from progenitor cells (pro-B cells) and pre-B cells, the cells undergo stringent selection at pre-BCR-checkpoints. A basal level of PI3K-Akt signaling at this level ensures differentiation between self and foreign antigens and eliminates auto-reactive clones via negative selection [90]. PTEN, as a classical TSG, prevents indiscriminate checkpoint activation by inhibiting PI3K-Akt signaling, preventing cell death. Thus, PTEN deletion causes suppression of pre-B-ALL development and regression of leukemia in vitro and in vivo. Similarly, in contrast to T-ALL patients where high PTEN expression predicts a favorable outcome, pre-B-All patients with high PTEN level show a poor prognosis. Contradictory findings have been reported in B-ALL where CK2α, frequently overexpressed and hyperactivated in all cancers including leukemia, phosphorylates PTEN at its C-terminal leading to its stabilization with simultaneous inhibition of its activity [91,92,93]. This is associated with PI3K-Akt pathway activation, commonly observed in adult B-cell ALL. These contradictory reports weaken PTEN’s significance as a therapeutic target. Findings of Shojaee et al. were corroborated by exposing pre-B-ALL cells to small molecule inhibitors of PTEN leading to hyperactivation of PI3K-Akt, phosphorylation, and activation of TSG p53, and cell death [89]. On the other hand, the latter report [91] used CK2 inhibitor CX-4945 that reversed PTEN level, and along with PI3K inhibitor LY294002 resulted in PI3K-Akt pathway inhibition and cell death. Thus, more work is required to distinguish the role of PTEN in pre-B-ALL cells vs adult B-ALL cells and in leukemia development.

## 5. FOXO Family

Forkhead box (FOX) proteins are evolutionarily conserved transcription factor family of proteins that regulate various cellular processes, such as apoptosis, cell cycle progression, and autophagy. Though typically considered as a tumor suppressor, a plethora of evidence substantiates the role of FOXO family members in positively regulating cellular proliferation, differentiation, and oncogenesis. These results might explain the unnatural overexpression of FOXO family proteins and ensuing poor prognosis and lower overall survival in multiple cancers, such as GBM, AML [94], breast cancer including triple negative breast cancer (TNBC), gastric and hepatocellular carcinoma, lymphoma [95], and invasive ductal carcinoma [96] of the breast.

FOXO family members might undergo deregulation or constitutive activation in human cancers via several mechanisms such as somatic alterations, chromosomal translocations, somatic point mutations, increased FOXO mRNA translation, altered PTM, or altered interactions with other transcription factors. Progressive research has shown that these mechanisms and interactions can promote cellular proliferation as well as cell death. In this review, we discuss a selection of these that incur oncogenic role on FOXOs.

### 5.1. FOXO’s Chromosomal Translocation in Cancer

The first reports of FOXO’s role in tumorigenesis came from the observations that FOXO factors are present at chromosomal breakpoints in human tumors. The chromosomal translocations lead to chimeric proteins, with the C-terminal domains of FOXO factors fused to N-terminal domains of other transcriptional regulators. FOXO1 and FOXO3/FOXO4 fuse with PAX3/PAX7 and Mixed Lineage Leukemia (MLL) gene in Alveolar Rhabdomyosarcoma (ARMS) and AML, t(X;11), ALL, t(6;11), respectively [97,98,99,100,101,102,103]. Two hypotheses have been proposed as explanations from the point of FOXOs GoF. ARMS tumors express a higher level of the fusion products, PAX3/FKHR and PAX7/FKHR, accompanied by a much higher transcriptional activation than the wild type PAX proteins. The GoF-FOXOs lead to oncogenic initiation and progression. This can also be explained as FOXOs LoF as disruption of one FOXO allele results in a partial loss of FOXO proteins and promotes cell cycle progression. 

FOXOs are rarely lost from chromosome or mutated in human cancers in a way that would render it completely inactive, since at least four FOXO alleles need to be inactivated for a complete FOXO LoF. However, single somatic mutations in FOXO have been reported to contribute to oncogenesis.

### 5.2. Oncogenic Role of Nuclear FOXO

FOXOs function downstream of oncogenic kinases, accumulate in the nucleus, and induce transcription of genes downstream of the signaling kinases, thus participating in feedback regulation within these pathways. FOXOs either directly activate the PI3K-Akt pathway upon treatment with doxorubicin [104], or regulate mTORC2 that activates Akt through Ser473 phosphorylation [105,106]. Furthermore, on PI3K-Akt inhibition by LY294002, activated nuclear FOXO3a enhances androgen-dependent Androgen Receptor (AR) expression and promoter activity, rescuing the cells from LY294002-induced apoptosis [107]. Treatment with PI3K/mTOR or Insulin pathway inhibitors reveal that FOXOs can upregulate the expression and activity of Receptor Tyrosine Kinases (RTK) such as ERBB2/3, IRS1/2, Insulin like growth factor receptors and RAS-MEK-ERK pathway [108,109]. These feedback mechanisms re-establish the proliferative signaling that builds up resistance against pathway-inhibitor drugs. 

Uncoupling of nuclear Akt-FOXO3a associated with predominant nuclear FOXO3a is also connected with poor prognosis, lymph node metastasis, and breast cancer progression [96,110]. Serum starvation, commonly associated with tumor progression, leads to nuclear retention of FOXO3a, due to deletion of serum factors such as IGF-1. Nuclear FOXO3a initiates transcriptional activation of MMP9 and MMP13 either by directly occupying their promoters or, as a co-activator, recruiting other transcriptional factors such as SMAD3/4 or NF-κB [111]. Furthermore, MMP induction is linked to breast cancer invasion and progression. In anaplastic thyroid carcinoma (ATC), nuclear FOXO3a promotes the proliferation of human ATC cells through transcriptional upregulation of cyclin A1 [112]. FOXOs also take part in tumor metastases in aggressive breast cancers, pancreatic ductal carcinoma, and GBM xenografts. Nuclear FOXO3a interacts with β-catenin, represses apoptosis, and promotes metastasis in colorectal carcinoma [113]. 

### 5.3. Foxo3a and p53

As bona fide TSGs, p53 or FOXO3a regulate a common pool of genes, and upregulation of either leads to cell cycle arrest and apoptosis. However, under specific cellular conditions, these proteins can evolve and adapt by changing their pattern of interaction. In response to serum starvation, p53 causes upregulation of cell survival factor SGK1 that phosphorylates FOXO3a and inhibits its transcriptional activity, protecting cells from ensuing apoptosis. SGK1 is also induced post-translationally via MAPK pathway in a p53-dependent manner and inhibits activation of FOXO3a, under conditions of genotoxic stress [114]. Under weak exposure to oxidative stress, p53 interacts with FOXO3a, inhibiting FOXO3a-induced expression of pro-apoptotic gene *BIM*. 

### 5.4. FOXO and Estrogen Receptor Alpha (ERα)

The role of FOXO3a in breast cancer progression remains controversial since multiple studies report contradictory and contextual interaction between nuclear receptors (NR) and FOXO3a. 

One of the earliest reports identified FKHR as either a co-activator or co-repressor of NR action, depending on the receptor type [115]. FKHR interacts with ER in HepG2 cells, causing a ligand-dependent repression of ER-mediated transcription via estrogen response element (ERE) sites, and inhibits the proliferation of ER-dependent MCF7 cells. In addition, FKHR also causes repression of transactivation by glucocorticoid receptor and progesterone receptor, while stimulating the actions of retinoic acid receptor and thyroid receptor.

In a contradictory report, in ERα^+^ MCF7 cells, interaction between FKHR and ERα in the presence of ligand E2 augments ERα-mediated transactivation through ERE sites and simultaneously represses FKHR-mediated transactivation *via* insulin response element sites [116]. These initial reports indicating both cell line and promoter specific interaction of FKHR and ERα were corroborated soon later [117].

Another factor determining the effect of FOXOs in breast cancer is the presence or absence of ERα [118]. Treatment with epigallocatechin-3-gallate (abundant in green tea) activates FOXO3a and ERα signaling and reverses TGFβ1-induced invasive phenotype in mouse mammary epithelial cells. This FOXO3a-mediated anti-epithelial mesenchymal transition (EMT) phenomenon is ERα-dependent and absent in ERα^–^ MDA-MB-231 cells [119]. In another study, a complex of FOXM1 and FOXO3a, occupying the identical FHRE elements on ERα promoter A, enhances FOXM1-dependent ERα transcriptional activity [120]. Oppositely, FOXO3a overexpression downregulates FOXM1 in MCF7 cells, where FOXO3a interacts with both ERα and ERβ through distinct domains, inhibiting their transcriptional activity [121]. It is postulated that FOXO3a-FOXM1 decreases FOXM1 expression, resulting a decrease in FOXM1-mediated ERα transcriptional activity, partially corroborating the previous reports. 

These contradictory reports demand further validation; however, holding out the fact that FOXO3a might cause increased ERα activity in context-dependent manner and attribute for breast cancer progression. 

### 5.5. Interactions with Other Oncogenes

An earlier study reports FKHR to act as a coactivator of STAT3 and modulate the IL-6-induced transcriptional activity [122]. FOXO3a can also interact with known oncogenic proteins that drive cellular proliferation. The Kaposi’s sarcoma-associated herpes virus latent protein LANA2 functionally interacts with FOXO3a, inhibiting the transactivation of FOXO3a target gene *Bim* in MCF7 cells [123]. Proto-oncogene *MYC*, binding with FOXO3a, represses FOXO3a-mediated activation of p27 promoter as corroborated from their inverse expression patterns in a diverse group of human cancers [124]. Oncogene FOXG1 prevents FOXO3a-p21 promoter binding, thereby enhancing tumor progression [125]. FOXO3a also promotes cellular invasion by BCL10-mediated overactivation of oncogenic NF-κB signaling [126]. 

### 5.6. TME and FOXOs

TME and FOXOs share a complex relationship, with FOXOs mediating the communication between the tumor cells and the immune system. As a cancer repressor, FOXO1 helps in T-cell trafficking, survival, and homeostasis. On the contrary, as a tumor promoter, FOXO1 and FOXO3 work in tandem in negatively regulating cytotoxicity of CD8+ T and NK cells against tumor cells [127]. A vast sea of literature exists that discusses the alternating roles of FOXOs in regulating both immune homeostasis and the immune response with respect to cancer immunity [127], further understanding of which is required to specifically target tumor/immune cells or both to optimize immunotherapeutic strategies.

In addition, hypoxia also maintains a relationship with FOXOs in that hypoxic microenvironment leads to FOXO3a upregulation, which via multiple pathways, including negative regulation of ROS levels, perturbs c-Myc dependent expression of mitochondrial genes. FOXO3a silencing in hypoxic tumor cells is associated with elevated ROS levels that sensitize cells to hypoxia induced cell death. Further, slowed tumor growth was observed in xenografts in vivo with increased caspase-3 staining indicating apoptosis. The tumors displayed increased expression of mitochondrial genes indicating increased mitochondrial metabolism [128]. 

On the other hand, hypoxia induced FOXO3a mediated ROS decrease further prevents Hif1a stabilization. Under chronic hypoxic conditions, FOXO3a mediated repression of mitochondrial gene expression and activity could rescue tumor cells from apoptosis and affect tumorigenesis and cancer progression [129,130].

### 5.7. Epigenetics and Maintaining Stem Cells Pool

Next to their primary role of tumor suppression and newly reported roles in promoting cell progression, FOXOs have been described to hold key importance in maintaining hematopoiesis. In the absence of FOXO3a, the self-renewal capacity of hematopoietic stem cells (HSCs) is abrogated, suggesting FOXO3a to be involved in the maintenance of the quiescent stage. FOXO3a^−/−^ HSCs show reduced expression of Catalase and *SOD2*, two important FOXO target genes involved in ROS detoxification, with the resultant ROS elevation disrupting the maintenance of HSC quiescence. Furthermore, elevated ROS leads to p38/MAPK activation in maintaining HSC’s stemness capacity [131]. In addition, in response to stress or insulin signaling, either by themselves or via interaction with chromatin modifiers such as CBP/p300, leading to alteration of histone acetylation/methylation at FOXO target genes, FOXO family members can alter the chromatin structure enhancing stem cell self-renewal and differentiation. They are found to complex with class III histone deacetylase (HDAC) SIRT1 deacetylates to increase cell quiescence and decrease apoptsis [132].

FOXO3a has a distinct opposing role on the survival of leukemia initiating cells (LICs) and non-LICs of CML. In non-LICs, BCR-ABL-driven strong Akt activation forcefully represses FOXO3a functions. In contrast, in LICs, active TGF-β signaling led to high SMAD 2/3, low Akt phosphorylation levels, and increased FOXO3a nuclear localization. This TGFβ-FOXO signaling is further responsible for the maintenance of CML LICs [133].

FOXOs are active in 40% of AML patient samples regardless of genetic subtype. FOXO1 is consistently upregulated in CD34^+^ t(8;21) AML cells, promoting a pre-leukemic state with enhanced self-renewal and dysregulated differentiation [134]. Disruption of all three functional FOXO genes, FOXO1, FOXO3, and FOXO4 led to a significant decrease in the population of long term HSCs in the quiescent stage, with more number of HSCs undergoing differentiation, thus underlining their role in preserving self-renewal capacity [135]. Separately, FOXO3a is known to suppress breast cancer stem cells (BCSC) properties and tumorigenicity via inhibition of oncogenic FOXMI/SOX2 signaling. In this context, DNA methyl transferase (DNMT1)-mediated FOXO3a promoter hypermethylation leads to downregulation of FOXO3a expression in BCa, leading to the maintenance of BCSC population and associated with development of drug resistance [136]. Further, FOXO3a in collaboration with another TSG liver kinase B1 (LKB1) regulate CD44 expression and play a central role in CSC maintenance in pancreatic ductal adenocarcinoma [137]

Another contrasting role of FOXOs are found in reducing sensitivity of BCa cells to HER2 inhibitor, lapatinib. Interestingly, lapatinib or any other PI3K-Akt inhibitor leads to stabilization of FOXOs in the nucleus, which recruit MLL2, a protein involved in H3K4 trimethylation along with histone acetyltransferase GCN5 on the promoter of oncogene *MYC*, leading to its acetylation and increased gene transcription. This MLL2/FOXO/c-Myc axis thus reduces sensitivity of BCa cells to Her2 inhibitors [138].

All these studies elucidate the fact that FOXOs partner with several epigenetic modifiers in collaboration with oncogenic signaling kinases that are responsible for stem cell maintenance and drug resistance/sensitivity. However, inhibitors of such epigenetic regulators and signaling pathways could not help to eliminate the tumors completely, suggesting possible role of additional microenvironmental factors in determining cell fate. Hence, additional efforts in understanding the several PTMs of FOXOs specific molecular backgrounds are required. 

### 5.8. Involvement in Stress Resistance and Longevity

Several studies have linked the role of FOXO family members to maintaining adult stem cell pools, longevity, and stem cell’s self renewal [132]. Growth factor deprivation, PI3K-Akt mutations, and stress conditions cause nuclear localization of FOXO family factors that upregulate a series of their target genes promoting stress resistance. For example, increased phosphorylation of FOXO4 at T451 and T447, by GTPase Ral-activated JNK mediates increased FOXO4 transcriptional activity. FOXO4-activated Catalase and SOD cause detoxification of ROS, thus alleviating oxidative stress. Since stress resistance is highly associated with longevity, activation of FOXO factors in the nucleus leads to extension of lifespan [139]. The cumulative role of FOXOs in stem cell maintenance, stress resistance, increased longevity, and life span can be manipulated to define an approach towards regenerative medicine.

## 6. PML

PML is the essential component of multiprotein sub-nuclear structures referred to as PML-Nuclear Bodies (PML-NBs), and is responsible for their formation, stability, and function. As a key regulator of many biological processes, PML multimerizes to form a scaffold for the assembly of proteins and transcription factors, with about 257 interactors governing more than 500 interactions (http://www.thebiogrid.org/). PML, through its transient or facultative interactions, facilitates tumor suppressive roles. We have demonstrated previously that PML plays pivotal role as part of various signaling pathways and as downstream of several transcription factors to regulate apoptosis and cell cycle arrest, and elicit anti-proliferative actions [140,141,142]. Though predominantly expressed in the nucleus, certain PML isoforms such as PML I and PML VII lack the nuclear localization/exclusion sequence and are potentially cytoplasmic (cPML). Additionally, two different PML mutations (1272delAG and IVS3–1G-A) lead to the formation of cPML [143]. These mutants show dominant negative effects over nuclear PML and inhibit p53-mediated cell growth suppression. Additionally, cPML sequesters and stabilizes PML-RARα in the cytoplasm, accentuating the latter’s oncogenic role [144]. A paradoxical role of cPML is observed pertaining TGFβ signaling. In one study cPML via association with SMAD 2/3 and SARA activates tumor-suppressive TGFβ signaling facilitating apoptosis [145]. In a paradoxical study in prostate cancer, cPML aided by the TGFβ receptors gets associated with SMAD 2/3 and SARA leading to their phosphorylation, nuclear translocation, and regulation of factors involved in promoting EMT and invasion [146]. This opposing function can be attributed to the cellular context of PML functioning, viz., a physiological model of study by Lin et al., and a clinical case of prostate cancer in the latter study. The results were corroborated with increased cPML expression in hepatocellular carcinoma as well. PTM of many TSGs takes place inside PML-NBs, dictating the ultimate mode of actions. However, newer pro-oncogenic roles of PML have been identified underlying potential therapeutic strategy.

### 6.1. Oncogenic Role of PML through PML-NB Interactions

In response to ultra-violet radiation, p53 and its regulators get sequestered inside PML-NBs where HIPK2-mediated phosphorylation and CBP/p300-mediated acetylation lead to increased p53 activation. However, under specific contexts, such as SIRT1 overexpression, p53 undergoes de-acetylation thus repressing its transactivation and PML-mediated cellular senescence [147]. PML also regulates the expression and activity of oncogenic factors by sequestering their co-regulators in the NBs. Upon genotoxic stress, IKKε and SUMO E3 ligase TOPORS get recruited inside the NBs and SUMOylated IKKε phosphorylates and activates p65 subunit of NF-κB contributing to its anti-apoptotic function [148]. 

Intriguingly, PML displays several anti-viral defense mechanisms that also play a critical role in virus-mediated oncogenic transformation. Interaction with PML isoforms IV/V and localization in PML-NBs allow SUMO modifications of human adenoviral oncoprotein E1B. PML-interacting E1B sequesters p53 in subnuclear aggregates, functions as SUMO E3 ligase, represses p53 transcriptional activity and p53-mediated apoptosis, facilitating efficient cellular transformation [149].

PML provides a docking platform for PPARγ coactivator 1A (PGC1α) and its regulators, notably SIRT1. Following SIRT1-mediated deacetylation and activation, PGC1α in PML-overexpressing breast cancer cells, particularly in TNBC and basal cells, promotes fatty acid oxidation (FAO) transcriptional program leading to increased ATP production that allows increased cell proliferation and migration thus providing a survival advantage [150]. Interestingly, the studied systems have inactivated p53. This deviation from PML-mediated p53 activation possibly further supports the tumor promoting metabolic role of PML associated with high tumor grades and poor prognosis in breast cancer.

mTOR inhibitor (rapamycin) and EGFR inhibitor (erlotinib) block downstream mTOR signaling, promote nuclear PML expression in GBM that targets and inhibits PI3K-Akt/mTOR signaling, slowing down cell proliferation. On the other hand, this cumulative effect overrides the growth inhibitory effects of mTOR inhibitors, thus providing resistance to mTOR and EGFR inhibitor treatment [151]. One possible explanation is that hampered kinase pathways lead to a slower cell cycle inducing a quiescent state, promoting resistance. However, PML’s metabolic pro-survival effects might also promote drug resistance in GBM, considering that GBMs with mutant EGFR rely on FAO for survival, with PML acting as interface. This study suggested the combinatorial approach of synergistic use of mTOR and EGFR inhibitors along with PML inhibitor arsenite (As_2_O_3_) for successful treatment of GBM patients.

### 6.2. Stem Cell Maintenance 

PML, located at chromosome 15, was originally identified to be involved in reversible chromosomal translocation with Retinoic Acid Receptor alpha (RARα) at chromosome 17, forming fusion oncoprotein PML-RARα which, acting as a dominant negative form of PML, abrogates PML’s tumor suppressive functions [152]. 

Of note, PML-RARα acts as a transcriptional repressor by allowing aberrant recruitment of epigenetic modifiers such as HDAC-containing complexes and DNMTs, Dnmt1 and Dnmt3a that leads to histone hypoacetylation and DNA methylation of PML-RAR target genes, such as a putative tumor suppressor *RARβ*, thus leading to their repression and silencing. Notably, treatment with pharmacological doses of all-trans retinoic acid reverses this hypermethylated phenotype, indicating APL to be a “treatable” form of leukemia [153].

Retinoic acid and As_2_O_3_ cause transcriptional repression of PML-RARα leading to the growth arrest of LICs in mouse APL [154], thus indicating a positive role of PML-RARα in maintaining cancer stem cell (CSCs) in leukemia. In HSCs and CML (an HSC disease) blasts, where the minor and rare sub-population of LICs are responsible for self-renewal and relapse, PML expression was seen to be elevated, with a strong negative association between PML positivity and clinical outcome. It was understood that PML indeed is responsible for HSCs and LICs maintenance and its As_2_O_3_-dependent downregulation is an effective approach for LICs eradication [155].

Further, PML-PPARδ-FAO pathway controls the HSCs asymmetric cell division and stem cell maintenance, thus linking cell metabolism and stem cell maintenance. Pharmacological inhibitors of FAO and PML, either alone or in combination, are proposed as therapeutic strategies to promote LIC exhaustion, thus targeting the proliferating leukemic cells [156]. Another study revealed that PML is essential for the maintenance of GBM-SCs and As_2_O_3_-induced rapid degradation of PML along with proto-oncogene c-Myc led to severe apoptosis [157]. As_2_O_3_ treatment also reduced the stemness of CD133^+^/CD13^+^ HSCs, an activity mediated by post-transcriptional suppression of PML and inhibition of Oct4, Sox2, and Klf4 expression [158]. 

Contradictory to its role in maintenance and proliferation of neuronal precursor cells (NPCs), LICs, and HSCs, PML inhibits proliferation of human mesenchymal stem cells (hMSCs); however, PML overexpression leads to CBP activation that allows pCREB-stimulation of *IBSP*gene, a marker for osteogenic differentiation [159]. A very recent report suggested that PML is functional in hMSCs by sustaining leukemic cells in a non-cell autonomous manner through the regulation of pro-inflammatory cytokines Cxcl1 and Il6 [160].

Apart from maintaining stem cell population, PML is also involved in regulating stem cell properties. Orphan nuclear receptor Tr2, localized in the PML-NBs, recruits co-activator PCAF in enhancing its target gene Oct4. ERK1/2-phosphorylated Tr2 is subjected to SUMOylation that releases Tr2 from the NBs and allows it to swap its co-activator with co-repressor Rip140, thus regulating Oct4 expression and stem cell proliferation [161,162,163]. A recent study illustrates that PML maintains naive pluripotent state in mouse embryonic stem cells (mESCs) and physically interacts with crucial regulators of pluripotency such as Oct4, Sox2, Nanog, c-Myc that contribute to self renewal [164]. As opposed to somatic differentiated cells, in mESCs, PML maintains pRb in a phosphorylated (inactive) state, while its loss impairs cell cycle progression and downregulation of critical factors responsible for cellular respiration and mitochondrial function, thus underlining the role of PML in the transition of mESCs to epiblast-derived stem cell state. Additionally, PML facilitates early EMT activation steps contributing to induced pluripotent stem cell generation. 

### 6.3. Epigenetics, Chromatin Association, and Transcriptional Control

Epigenetics and PML-NBs work in a closed-circuit loop [165]. First, several epigenetic regulators such as histone methyltransferases/deacetylases or DNMTs and chromatin remodeling enzymes, transcriptional co-activators, and co-repressors have been identified as components of PML-NBs, indicating the role of PML-NBs in epigenetic regulation. Second, despite the absence of a canonical DNA binding site, PML largely assembles in euchromatin regions, modulates the activity of transcription factors, can be recruited to target gene promoters, as revealed by chromatin immunoprecipitation studies [166,167]. Finally, stable PML bodies contribute to genetic stability allowing the conservation of the genetic and epigenetic gatekeepers that sense DNA damage and participate in DNA repair. In this context, aberrant epigenetic regulation could destabilize PML-NBs and compromise their association with specific genetic loci, coupled with compromised ability to sense DNA damage, which in turn makes PML-NBs unable to maintain epigenetic regulation, all defining steps of tumorigenesis. As examples, PML functionally associates with Fos and Jun and regulate AP-1-mediated gene transcription, specific targets of which have not been carefully studied so far [168]. PML also associates with c-Jun upon UV radiation, promotes its transcriptional activity and induces apoptosis [169]. Additionally, PML4 interacts with SP1, promotes its SUMOylation and sequestration inside the PML-NBs, further reducing SP1-binding to its target genes Survivin and EGFR, inhibiting SP1-mediated gene transcription [170,171]. This might explain the PML-mediated downregulation of Survivin promoter observed in osteosarcoma cells [172]. PML also colocalizes with β-catenin and p300, affects β-catenin-mediated transcription and upregulation of tumor suppressor ARF, but not cyclin D1 [173]. Coincidentally, all the reports here point to PML-mediated repression of oncogenic promoters or activation of mediators of apoptosis. Further study and analysis of several other target genes, viz., AP1/SP1/TCF-LEF might point to a contradictory pro-oncogenic role of PML in gene regulation.

In accordance with the pro-survival role of PML in breast cancer, PML is upregulated in TNBC, a subset of breast cancers. PML is a direct HIF-1α target gene, its expression correlating with other pro-metastatic hypoxia signature genes. PML binds to the regulatory regions of *WIPF1, PLOD1*, and *ZEB2* genes and participates in regulation of HIF-1α-dependent metastasis [174].

The contribution of PML in maintaining stemness and breast cancer metastasis provides a promising strategy in targeting PML towards a more context-dependent effective therapy.

## 7. Conclusions

It is now well established that cancer is fundamentally a complex and heterogeneous genomic disease. The Cancer Genome Atlas (TCGA) Research Network has analyzed and profiled a wide variety of cancers and identified genetic and epigenetic alterations that can promote oncogenesis [175]. Several transformation assays, genetic linkage analyses, copy number variation studies, and bioinformatics analyses have helped in identifying and differentiating the oncogenes and TSGs [176,177,178,179]. Interestingly, certain TSGs possess both oncogenic and tumor suppressive functions. It should not be a surprising event, considering the fact that these major TSGs are involved in an array of fundamental cellular signaling (Figure 4), and thus their aberrant roles might arise from different cellular contexts, resulting in a wide range of pathological outcomes. The wild type forms of these genes are poorly expressed in cancers of different histological origins, barring leukemia, where they are overexpressed, with a prominent role in maintaining the LICs. This indeed is one commonality in function, where in various capacities they help in maintaining CSCs and upregulating CSC-specific markers and transcription factors. The TSGs mentioned here are closely connected by two distinct pathways: p53 and PI3K-Akt signaling and are often in close proximity to PML-NBs. The detailed mechanistic explanation of the duality of action in each associated cancer type might seem like an impossible task, but it is worth investigating. Well-designed bioinformatics analyses, similar to the one conducted by Shen et al. [5], can help in elucidating the functional duality and the physiological balance maintained at different cellular contexts.

The large number of examples, as illustrated in this review and elsewhere, suggests that the function of a gene cannot be guaranteed in terms of standard definitions. A review by Paige, discussing the exceptions to the two-hit hypothesis, including a multi-step cancer progression, highlights this concern [7]. As rightly stated therein, a TSG should not be “assessed solely on a correlation between mutation and cancer”. Despite the fact that TSGs are found to be altered more often than oncogenes in human cancers [180], most of the current molecular therapies aim at targeting oncogenes, by developing inhibitors against them. Exploiting TSGs can be helpful in sharing the therapeutic load. However, TSGs not being cell surface receptors or rarely being kinases, developing therapeutic strategies against them remain challenging. Additionally, the existence of dual roles opens up newer opportunities and renders the approach more complex. The functional components of each TSG and their roles in different cancer types and individual cancer cases need to be addressed. Accordingly, therapeutics based on standard TSG norms need to be re-validated. A primary step in this regard would be to question the norms, understand the mechanism of action, and report the findings, which would help in addressing the evolutionary changes in cancer genetics.

## Figures and Tables

**Figure 1 cells-10-00046-f001:**
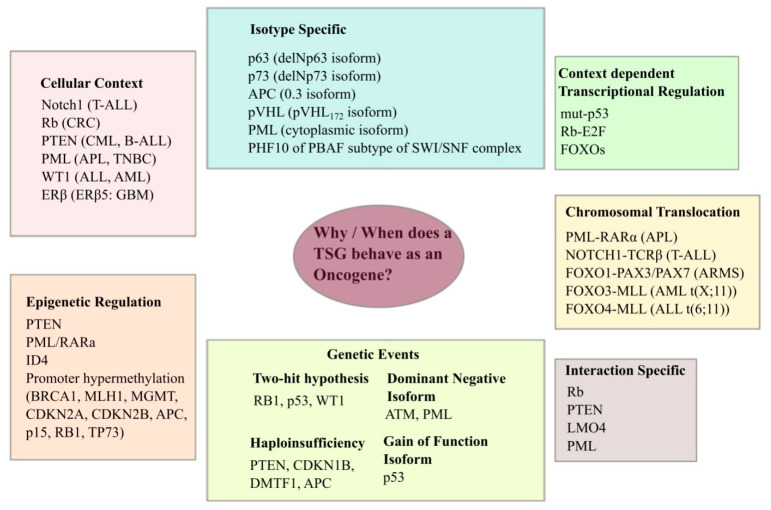
Deviation of tumor suppressor genes from displaying typical tumor-suppressive behavior: Identification of possible situations with examples of candidate Tumor Suppressor Genes (TSGs). (Modified and re-adapted from Paige AJW. *Cell Mol Life Sci* 2003; **60**: 2147–2163.) [7].

**Figure 2 cells-10-00046-f002:**
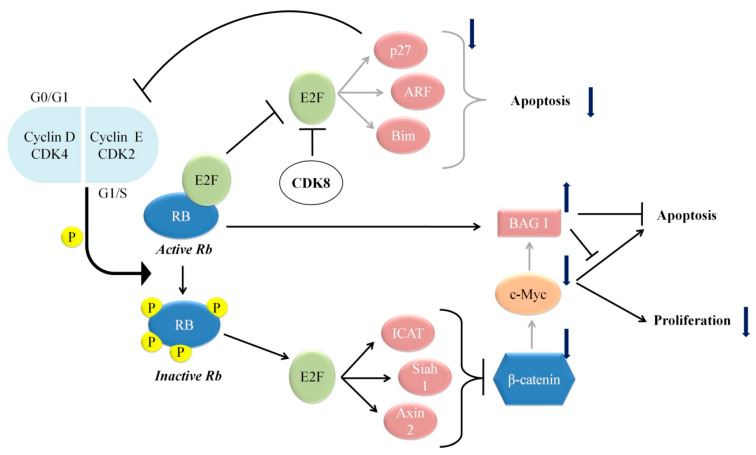
Schematic representation of role of Rb-E2F signaling in cellular proliferation.

**Figure 3 cells-10-00046-f003:**
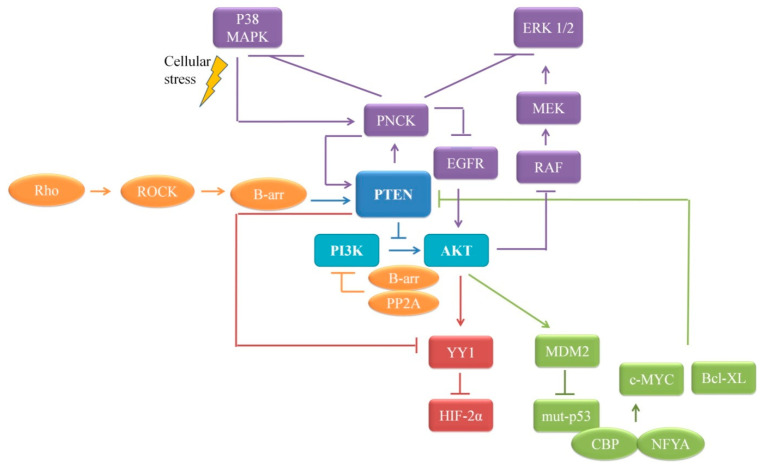
Schematic representation of the role of PTEN and PI3-Akt signaling in cellular proliferation.

**Figure 4 cells-10-00046-f004:**
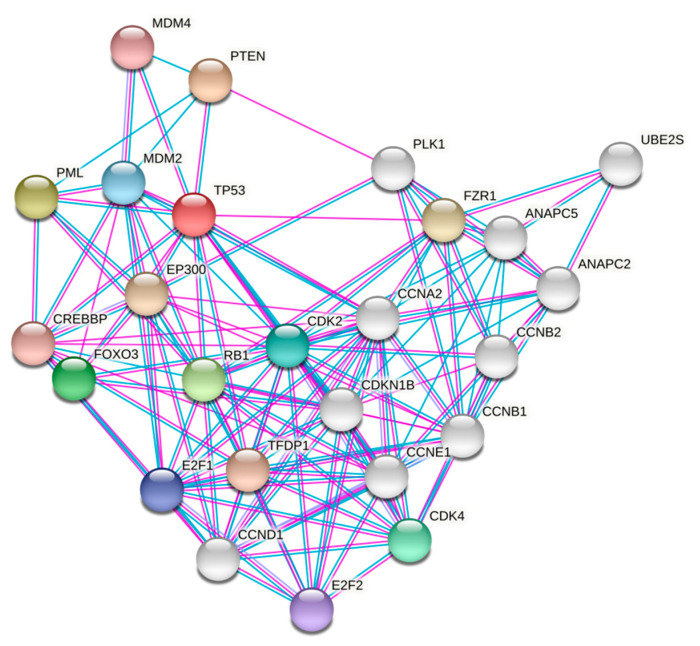
A STRING (https://string-db.org/) summary showing known protein–protein interactions of reviewed tumor suppressor proteins. The nodes represent interacting proteins and the edges represent the interactions/associations. The blue lines represent interactions studied from curated databases and purple lines represent experimentally determined interactions. Colored nodes are the query proteins and their first shell of interactors and the empty of grey nodes represent second shell of interactors.

**Table 1 cells-10-00046-t001:** List of tumor suppressor genes (TSGs) with potential oncogenic role (adapted from TSG2.0).

Classification		Genes
Protein-coding	Transcription factors	*FOXL2, RUNX1, DNMT1, DNMT3A, ETS2, ETV6, EZH2, FOXO1, FOXO3, GLI1, HDAC1, FOXO4, MXI1, NOTCH1, NOTCH2, NOTCH3, PAX5, RARB, SKIL, TCF3, WT1, ZBTB16, NR4A3, NCOA4, KLF4, LITAF, YAP1, SALL4, HOPX, LHX4, FUS*
	Kinases	*BCR, CDKN1B, MAP3K8, FLT3*
	Protein binding	*RHOA, ECT2, IDH1, NPM1, PHB, PML, PTPN11, SPOP, RASSF1, ARHGEF12, SIRT1, SUZ12, WHSC1L1, WDR11, RB1, CBL, DMBT1*
Noncoding RNA (ncRNA)		*MIR106A, MIR107, MIR125B1, MIR146A, MIR150, MIR155, MIR17, MIR18A, MIR194-1, MIR194-2, MIR196A2, MIR20A, MIR203A, MIR210, MIR214, MIR222, MIR223, MIR24-1, MIR27A, MIR18B*

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
