# Peer review of "Tumor Suppressors Having Oncogenic Functions: The Double Agents"

_cells, 2020, doi:10.3390/cells10010046_

Round 1

Reviewer 1 Report

In this manuscript titled “Tumor suppressors having oncogenic functions: the double agents” Datta et al reviewed emerging paradoxical roles of key cancer related proteins. By definition tumor suppressor proteins are usually anti-proliferative in nature and gets inactivating mutations in the course of tumor initiation. Similarly oncogenes are proliferative by nature and they get constitutively active during tumorigenesis. However, many solid reports in last decade established the opposing role of key tumor suppressors and oncogenes. Usually these contradictory roles are rare and context dependent. In this manuscript the authors carefully reviewed our current understanding about these paradoxical roles.   Authors have meticulously discussed this phenomenon by taking the examples of pRb, PTEN, FOXO and PML. By taking the examples of these genes authors discussed that extensive research over the last few years have indicated that certain tumor suppressor genes do not conform to these standard definitions and might act as ‘double agents’, playing contrasting roles in vivo in cells, either due to haploinsufficiency, epigenetic modifications or due to involvement with multiple genetic and oncogenic events. In my knowledge this is one of the first manuscript to comprehensively discuss these emerging phenomenon. My specific comments are as follows:

  1. Although authors have mentioned about role of epigenetic factors in reversing the nature tumor suppressors, they did not discuss it in detail in the main text. I suggest them specifically mentioned about epigenetic and micro-environmental factors in reversing the nature of TSGs.
  2. Are there some specific mutational signatures of the mentioned TSGs which makes them oncogenic?

Author Response

Reviewers’ Comments:

Reviewer #1 (Comments and suggestions to the Author): 

In this manuscript titled “Tumor suppressors having oncogenic functions: the double agents” Datta et al reviewed emerging paradoxical roles of key cancer related proteins. By definition tumor suppressor proteins are usually anti-proliferative in nature and gets inactivating mutations in the course of tumor initiation. Similarly, oncogenes are proliferative by nature and they get constitutively active during tumorigenesis. However, many solid reports in last decade established the opposing role of key tumor suppressors and oncogenes. Usually these contradictory roles are rare and context dependent. In this manuscript the authors carefully reviewed our current understanding about these paradoxical roles. Authors have meticulously discussed this phenomenon by taking the examples of pRb, PTEN, FOXO and PML. By taking the examples of these genes authors discussed that extensive research over the last few years have indicated that certain tumor suppressor genes do not conform to these standard definitions and might act as ‘double agents’, playing contrasting roles in vivo in cells, either due to haploinsufficiency, epigenetic modifications or due to involvement with multiple genetic and oncogenic events. In my knowledge this is one of the first manuscript to comprehensively discuss these emerging phenomena. My specific comments are as follows:

  1.  authors have mentioned about role of epigenetic factors in reversing the nature tumor suppressors, they did not discuss it in detail in the main text. I suggest them specifically mentioned about epigenetic and micro-environmental factors in reversing the nature of TSGs.
  2. Are there some specific mutational signatures of the mentioned TSGs which makes them oncogenic?

We extend our many thanks to the reviewer for the in-depth analysis, the appreciative words, and the suggestions that have helped in improving the manuscript. We have tried to address all the concerns that have been raised. We have added point-to-point responses as follows.

  1. We earnestly thank the reviewer for the critical comments. Yes, we realize we had not discussed much about epigenetic regulation and microenvironmental factors in our original manuscript. A partial reason behind that would be, most commonly, any sort of promoter hypermethylation (the predominant form of epigenetic regulation) leads to silencing of the target genes, and in this case, it would automatically cause repression of the TSGs, leading to an oncogenic impact. This is a natural course of action.

However, keeping in mind your suggestion and the fact that prominent epigenetic regulations do demand a mention, when we are discussing the conditions where TSGs turn oncogenic, we have incorporated several facts and references in due places in the revised manuscript.

We thank the reviewer for the mention of inclusion of microenvironmental factors, since there are indeed several instances where the tumor microenvironment directs the TSGs in reversing their roles. TSGs most often work in tandem with the microenvironment to determine the cell fate. Since there is a huge amount of literature that discusses TME and TSG, we have incorporated the major ones, to keep the word count in check, having cited relevant references.

Lastly, we have modified and edited the manuscript, and have tried to bring about clarity for an easy and better read. We hope this manuscript is now suitable for publication in Cells.

Here are the changes being incorporated:

 pRb

Role of tumor microenvironment: Page 5 (lines 150-153).

Role of epigenetic factors: Page 4 (lines 121–125); Page 5 (lines 145–147).

PTEN

Role of tumor microenvironment: Page 8 (lines 281–285); Section 4.6 Page 9 (lines 328–336).

Role of epigenetic factors: Section 4.2 Page 7–8 (lines 256–263)

FOXO Family

Role of tumor microenvironment: Section 5.6 Page 12–13 (lines 465–485).

Role of epigenetic factors: Page 13 (lines 493–498; 509–527).

PML

Role of tumor microenvironment: Page 14 (lines 548–561)

Role of epigenetic factors: Page 15 (lines 600–605), Page 16 (lines 639–642; 645–650).

All the additions have been made in track changes for your perusal.

  1. We thank the reviewer for this question.

TSG inactivation is primarily attributed to several genetic changes, namely, microdeletions that were identified by their demonstration of loss-of-function activity[1]. As we have mentioned in our opening section (Page 3-lines 87-103), the TSGs demonstrate several modes of mutational inactivation that render them oncogenic. It is natural of any TSG that harbors mutations/mutational signatures to turn their back on tumor suppression and be oncogenic in their function. There has been immense research that has been performed, reported, and conducted till date that tries to decipher the mutational signatures of TSGs and their role in cancer[2,3].

For example, PTEN is a classic example of a tumor suppressor, of which hundreds of germline and sporadic mutations have been reported till date[4,5]. Point mutations in PTEN are most commonly detected in CNS, endometrium, prostrate, and the skin. Since the first discovery of autosomal dominant mutations in PTEN to be associated with Cowden Syndrome, in 1997, current research has further demonstrated evidence that links PTEN, Cowden Syndrome and meningioma in a string[6].

pRb, similar to PTEN, behaves as a classic TSG, whereby, inheritance of a germline mutation in one allele of the pRB-encoding gene is associated with the retinoblastoma disease. RB1 mutations are often found in astrocytomas and glioblastomas, though primarily, pRb loss is also associated with loss/inactivation of its family members, or being inactivated by Cyclin D/CDK4/CDK6-induced phosphorylation[7,8].

In contrast, mutations in FOXO are fewer in number. The most common mutational signature of FOXO is its participation in chromosomal translocations in alveolar Rhabdomyosarcoma, as mentioned in the manuscript. Some other instances would be a common chromosomal 6q21 deletion in mature B cell lymphomas and childhood acute lymphoblastic leukemia associated with FOXOX3a and a loss of FOXO1 in a chromosomal deletion at 13q14 associated with tumorigenesis of the benign mammary and vaginal myofibroblastomas[9–11]. However, FOXO genes are rarely mutated in cancer, suggesting some other mode of deregulated expression.

The most common mutational signature of PML would be its involvement in chromosomal translocation with RARα in AML, as already mentioned in the manuscript.

Having said that, it is a natural progression for any TSG to behave such a way in event of any mutation. However, in our review, we tried to focus on the ‘uncommon’ conditions/factors, where, in spite of having two functional alleles, or under constitutive activation, certain TSGs demonstrate an opposing phenomenon, a behavior opposite to their nature under natural conditions. In this context, we did mention incidences of chromosomal translocations, haploinsufficiencies, and epigenetic modifications that dictate TSG fate. We believe, however, that mentioning regular mutational signatures of these TSGs would a) not be in line with the main theme of the review and b) be too elaborate and beyond the scope of this review. We sincerely hope you understand the predicament and advise as you deem fit.

Reviewer 2 Report

COMMENTS TO THE AUTHORS

The authors have reviewed in this manuscript the current knowledge about the role of the tumor suppressors having oncogenic functions. The authors intended to provide insights into the controversial function of four relevant tumor suppressor genes (Rb, PTEN, FOXO and PML) that appeared to act also as oncogenes, in specific genetic and/or cellular contexts. In particular, the aim of the study is, within reason, to review systematically the current knowledge on the “double agents” issue, the ultimate objective being to evaluate the potential of such factors as therapeutic targets for drugs directed against different malignancies. For this reason, a Review on the subject might be of great interest.

General comments

The manuscript appears to be an extensive and well-conceived study reviewing the available knowledge on the matter by the cellular point of view, although the Authors provide also an integrated view regarding potential targeting therapies.

Author Response

Reviewers’ Comments:
Reviewer #2 (Comments and suggestions to the Author):

The authors have reviewed in this manuscript the current knowledge about the role of the tumor suppressors having oncogenic functions. The authors intended to provide insights into the controversial function of four relevant tumor suppressor genes (Rb, PTEN, FOXO and PML) that appeared to act also as oncogenes, in specific genetic and/or cellular contexts. In particular, the aim of the study is, within reason, to review systematically the current knowledge on the “double agents” issue, the ultimate objective being to evaluate the potential of such factors as therapeutic targets for drugs directed against different malignancies. For this reason, a Review on the subject might be of great interest.

General comments: The manuscript appears to be an extensive and well-conceived study reviewing the available knowledge on the matter by the cellular point of view, although the Authors provide also an integrated view regarding potential targeting therapies.

Ans: We earnestly thank the Reviewer for the analysis and the appreciative comments. We have modified and edited the manuscript, and have tried to bring about clarity for an easy and better read. We hope this manuscript is now suitable for publication in Cells.